# Aspects of Tertiary Prevention in Patients with Primary Open Angle Glaucoma

**DOI:** 10.3390/jpm11090830

**Published:** 2021-08-24

**Authors:** Gabriel Zeno Munteanu, Zeno Virgiliu Ioan Munteanu, George Roiu, Cristian Marius Daina, Raluca Moraru, Liviu Moraru, Cristian Trambitas, Dana Badau, Lucia Georgeta Daina

**Affiliations:** 1Faculty of Medicine and Pharmacy, University of Oradea, 410087 Oradea, Romania; gabi_munteanu_91@yahoo.com (G.Z.M.); zenomunteanu@yahoo.com (Z.V.I.M.); roiug70@yahoo.com (G.R.); cristi_daina@yahoo.co.uk (C.M.D.); lucidaina@gmail.com (L.G.D.); 2Faculty of Medicine, George Emil Palade University of Medicine, Pharmacy, Sciences and Technology, 540142 Targu Mures, Romania; raluca.moraru@umfst.ro (R.M.); cristian.trambitas@umfst.ro (C.T.); 3Petru Maior Faculty of Sciences and Letters, George Emil Palade University of Medicine, Pharmacy, Sciences and Technology, 540142 Targu Mures, Romania; 4Faculty of Physical Education and Mountain Sports, Interdisciplinary Doctoral School, Transilvania University, 500068 Brasov, Romania

**Keywords:** open angle primitive glaucoma, tertiary prevention, visual field (VF), routine health examination, intraocular pressure (IOP), screening

## Abstract

The purpose of the study is to assess the health of patients in the activity of tertiary prevention dedicated to preventing blindness caused by POAG (primary glaucoma with open angle and high tension) and NTG (primary glaucoma with open-angle and statistically normal tension—particular form of glaucoma with open angle) and preservation of the remaining visual function. The design of the study is epidemiological, observational, descriptive and retrospective, and uses only the data recorded in the existing records in the archives of the Ophthalmology office within the Integrated Outpatient Clinic of the Emergency Clinical Hospital of Oradea (IOCECHO) during the years 1999–2019 (anamnestic data; objective examination and paraclinical examination: intraocular pressure—IOP and visual field—VF). The methods of the study included the standardized protocol: anamnesis, physical ophthalmological examination, IOP determination, and computerized perimetry with the “Fast Threshold” strategy performed with the “Opto AP-300” perimeter. The obtained results were statistically processed with a specialized software (S.P.S.S.—I.B.M. Statistics version 22). The study examined the available data of 522 patients of which 140 were men (26.8%) and 382 were women (73.2%). The gender ratio was 0.37. In the period 1999–2019, 150,844 people with ophthalmic pathology were consulted in the Ophthalmology office of IOCECHO out of which 522 patients (0.35%) were diagnosed with primitive open-angle glaucoma, 184 people (35.2%) presented high IOP (POAG), and 338 people (64.8%) had statistically normal IOP (NTG). The annual proportion of cases diagnosed with glaucoma in the total number of patients examined was between 0.1% (2005; 2008; 2010) and 2.4% in 2012, when 101 people were detected. In the studied records, no cases of uni- and/or bilateral blindness were mentioned. The mean age of glaucoma patients at the first consultation was 60.81 ± 12.14 years with high frequencies in the 55–69 age groups and at the last consultation it was 66.10 ± 12.47 years with high frequencies in the age groups between 60–74 years. Monitoring and treatment of glaucoma patients was beneficial; IOP decreased statistically significantly: in patients with POAG by 46.16%, from 30.50 ± 7.98 mmHg to 16.42 ± 3.01 mmHg (*p* = 0.000) and in those with NTG by 17.44%, at 16.39 ± 3.66 mmHg at 13.53 ± 1.92 mmHG (*p* = 0.000). The duration of treatment and monitoring was on average 5.1 ± 3.4 years, for 184 patients (35.2%) with POAG and 5.1 ± 3.8 years for 338 patients (64.8%) with NTG. Tertiary prevention of glaucoma, by providing specialized care, ensures effective control of IOP and implicitly of the long-term evolution of the disease. IOP is the only modifiable risk factor in patients with POAG and NTG and its decrease prevents the progression of the disease and emphasizes the importance of early diagnosis and treatment. The management of the glaucoma patient consisted of: complete ophthalmological examination (subjective and objective), paraclinical examination with IOP, and VF measurement (valuable ophthalmological diagnostic tool) for disease detection and effective assessment of disease progression in order to improve the process of therapeutic decision making.

## 1. Introduction

Glaucoma is a multifactorial degenerative disease of the optic nerve, characterized by the progressive destruction of nerve fibers responsible for transmitting information from the eyes to the brain. This causes a gradual narrowing of the visual field and eventually blindness can occur [1,2]. Globally, glaucoma is the second leading cause of blindness and the first cause of irreversible (permanent) blindness [3,4,5]. There are several types of glaucoma: POAG (primary open-angle glaucoma with high intra-ocular pressure) is the most common form of glaucoma, which is responsible for over 90% of cases; NTG (normal tension glaucoma -primary open-angle glaucoma with normal intra-ocular pressure, particular form of open-angle glaucoma); PACG (primary closed-angle glaucoma); congenital glaucoma; and glaucoma secondary to other conditions (local trauma, adverse effects following treatments, etc.).

The prevalence of POAG is high, with a high blindness rate of 2% in the Caucasian population over 40 years. Estimates showed for 2020 a number of approximately 60–75 million patients with POAG, and for 6 million of them it was expected that they will evolve towards bilateral blindness. Subsequently, the prevalence estimates for POAG mention 3.5% in the general population in the 40–80 age group and 0.50% for PACG. The prevalence of glaucoma is influenced by race: POAG is more prevalent in people of color while PACG has an increased prevalence in East Asian populations. POAG is the most common clinical form while blindness occurs more frequently in PACG [6].

The risk of unilateral blindness in treated POAG is 15% at 15 years, and risk of bilateral blindness in treated POAG is 6% at 15 years. The incidence of blindness varies, being conditioned by: the time of specifying the diagnosis, the quality of treatment and monitoring of the disease, low compliance, and increased life expectancy. Early diagnosis and appropriate treatment reduce the rate of blindness in glaucoma [7]. Globally, an increase in the number of glaucoma patients to 111.8 million is estimated for 2040 of which 10% are already in the stage of bilateral blindness [8,9]. NTG is a progressive optic neuropathy with statistical IOP considered normal (normal statistical range ≤21 mmHg). The prevalence of NTG varies greatly between different population studies, as it is the most prevalent subtype of open-angle glaucoma. The etiology of GNT is not yet well known, but is possibly multifactorial [10]. There are no official statistics in Romania, but extrapolating the data from the European level, the number of glaucoma patients is estimated at 140,000 people, of which 132,000 are diagnosed with POAG. The etiology of the disease is not fully established, the disease is often associated with increased intraocular pressure (over 21 mmHg). The diagnosis is made by the ophthalmologist through specialized procedures and investigations.

The specific treatment does not cure the disease, as vision loss cannot be recovered. Medicines used for treatment stop and/or delay the worsening of the disease, act mainly by lowering intraocular pressure [2]. Prevention basically means the purposes of medicine: promotion, preservation, restoration of health, and reduction of suffering and pain [11]. For glaucoma patients, secondary prevention is a way of detecting the disease through specific screening activities and regular medical control [12]. “Screening involves the fact that in a population there are unknown diseases and patients due to unmet, unexpressed or unsatisfied needs.” Selective or targeted screening is the most effective and efficient means of detection and can address people exposed to risk factors. The evaluation of the screening can address the whole process (early detection and intervention) or only the level of the treatment effect in those in whom the detection result was positive. The benefits of screening are greater for those at increased risk of developing the disease [12,13].

Tertiary prevention aims to prevent complications and aggravation of diseases. These are achieved through clinical and therapeutic activities that aim to reduce the evolution and complications of a disease, the suffering caused by deviations from health, injuries, and infirmities. Tertiary prevention is done after the onset of the disease, closely intertwined with the treatment of chronic disease. The effectiveness of preventive interventions is reflected by their ability to achieve the desired result in health (Health Effect) [14,15].

The retina is a layered structure with ten distinct layers of neurons interconnected by synapses. The cells subdivide into three basic cell types: photoreceptor cells, neuronal cells, and glial cells. Retinal Layers from outside in: (1) retinal pigment epithelium; (2) rods and cones (photoreceptors); (3) external limiting membrane; (4) outer nuclear layer; (5) outer plexiform layer; (6) inner nuclear layer; (7) inner plexiform layer; (8) ganglion cell layer; (9) nerve fiber layer; and (10) inner limiting membrane [16,17]. Ganglion cells collect information from bipolar cells and amacrine cells. This information is in the form of chemical messages sensed by receptors on the ganglion cell membrane. Ganglion cells group together and form the axons that become the optic nerve fibers. Nerve fibers within the retina send electrical signals to the brain, which then interprets these signals as visual images [18,19]. Glaucoma is a group of diseases characterized by progressive optic nerve degeneration that results in visual field loss and irreversible blindness. A crucial element in the pathophysiology of all forms of glaucoma is the death of retinal ganglion cells [20].

This is a descriptive work based only on documents available to the reference population (registered patients and treatment) and presents: the description and distribution of disease parameters and some risk factors according to different characteristics (personal, temporal, spatial and of the parameters of clinical, paraclinical investigations and of the applied treatment). The objectives of the study were to evaluate the epidemiological, clinical, and statistical evaluation of demographic, clinical, and paraclinical aspects (parameters of visual function) of patients with POAG and NTG from the records of the Ophthalmology office within the Integrated Outpatient Clinic of the Emergency Clinical Hospital) during the years 1999–2019. 

The present paper aims to describe the state of health and to investigate the possible relationships between risk factors and disease for patients with POAG and NTG in the records of the ophthalmology office of IOCECHO, in the activities of dispensary (monitoring) and treatment. 

## 2. Materials and Methods

### 2.1. Ethical and Legal Aspects

The study was carried out after obtaining the Ethical Council Approval (Document no. 8630/03.04.2019) and Ethics Commission Approval (Document no. 8805/04.04.2019) as well as unrestricted access to the archived data of patients for scientific research purposes (FOCG) within Oradea County Emergency Hospital. In addition, further approval was obtained from the Public Health Directorate of Bihor County in order to have access to specific statistical data performed within this institution (Approval no. 4439/11.03.2019).

### 2.2. Data Collection

The database was prepared in the period March–April 2020 by strictly recording the medical information entered in the individual monitoring sheet of the glaucoma patients and the results of the examinations of the edited visual field.

### 2.3. Study Design

The evaluation of the health status of patients with POAG was performed through a descriptive, retrospective study. The documents used to extract this data were from the databases available in the Ophthalmology office within the Integrated Specialist Outpatient Clinic of the Oradea County Emergency Clinical Hospital. The timeframe spanned over a period of 20 years, between 1999–2019. The specific statistical analysis was provided by the Public Health Directorate of Bihor County, Biostatistics Department and Public Health Informatics.

### 2.4. Methodology

The ophthalmology office within IOCECHO carried out permanent activities for active detection of glaucoma patients and in campaigns organized by the “Romanian Glaucoma Society” campaigns that were promoted in the local media on the occasion of “World Glaucoma Week” [2]. The study included patients with glaucoma, respectively with the two nosological entities: POAG (primary open-angle glaucoma with high intra-ocular pressure) and NTG (primary open-angle glaucoma with normal intra-ocular pressure), diagnosed, treated, and monitored, following the objective ophthalmological examination and the ocular functional examination. Two nosological entities were identified within the POAG: POAG (IOP ≥ 21 mmHg) and NTG (IOP ≤ 20 mmHg) [3]. NTG is a progressive optic neuropathy with IOP in the normal statistical range (≤21 mmHg). The prevalence of NTG varies greatly between different population studies and is the most prevalent subtype of open-angle glaucoma in some reports. The possible etiology of NTG would be multifactorial, but not yet well defined [21].

The exclusion criteria targeted other forms of open-angle glaucoma: primitive juvenile glaucoma, secondary glaucoma (pseudo-exfoliative, pigmented, with crystalline particles, associated with intraocular, uveitic, neovascular, associated with intraocular tumors, associated with retinal detachment, post-traumatic corticosteroid-induced, and surgical and/or laser treatment). Other eye diseases such as corneal, lens, vitreous, and retinal diseases, etc. were excluded.

Epidemiological, demographic, and specialized ophthalmological parameters were used to characterize the health status of patients with POAG. The epidemiological parameters were: number of disease cases, annual proportion of cases diagnosed with POAG in the total number of patients examined at the ophthalmology department, age, gender, sex ratio, place of residence, age of detection, age of last consultation, age of cessation of surveillance, and duration of monitoring and treatment.

The data of the considered medical interrogation were the anamnestic ones, the heredocolateral antecedents, the personal pathological antecedents and of the associated diseases. The objective ocular examination consisted of the determination and recording of IOP with Goldmann aplanotonometer and of the cup/disc ratio by direct ophthalmoscopy. IOP was considered an important indicator both for the detection of glaucomatous disease and in monitoring its progression under treatment. The ocular functional examination consisted of the determination of visual acuity and the determination of the visual field. Visual acuity was investigated with the Snellen optotype.

The determination of the visual field was done with the perimeter: “Opto AP 300—Computerized perimeter”, with the strategy “Fast Threshold” using optical correction as needed. The following parameters were considered: credibility indices (monitoring/loss of fixation, false positive answers, false negative answers), time required to perform the test, visual slope sensitivity (isopter level 3 degrees), theoretical slope of the visual slope to 10 degrees (slope), zero level, the average value, the structural defect (Pattern Defect), the average defect (Average Defect), and the graph of the defect (“Bebie Curve”). For the statistical interpretation of the graph of the centralized defect of a test result (“Bebie Curve”), we used the following categorical classification (Table 1) [22].

In the study, the cases were admitted according to: credibility indices, establishing a percentage threshold with additional qualitative descriptions (the proportion of 16–20% was considered at average for loss of fixation and maximum 15% for false positive and false negative errors).

### 2.5. Statistical Analysis

The analysis of the indicators of the descriptive statistics includes: the indicators of the central tendency (average), the indicators of the dispersion (standard deviation). The analysis of the distribution of the variables was performed with the Kolmogorov–Smirnov test. For the descriptive analysis to identify the potential differences between the groups of variables studied, the first and last consultation used non-parametric tests: Two-Related Samples Wilcoxon test for related scores. With the Binominal Test we compared a proportion with a specified value [23].

The correlation analysis studied the intensity of the links between the variables according to their distribution and was expressed by the correlation coefficients (numerical indices of the power and direction of the relationships between the variables). In the case of non-normal distributions, the Spearman’s rho coefficient was used, which in most cases remains unaffected by aberrant values. The degree of association was assessed depending on the value of the correlation coefficient as: strong (0.8–1), moderate (0.5–0.8), weak (0.2–0.5), and negligible (0–0.2). The evaluation of the intervention of the event is done by testing the statistical significance. When the variables were of the score type, the appropriate method of graphical analysis was the scatter plot. For the analysis of the nominal variables, the Chi-Square Test was used to evaluate the existence of a significant difference between two or more samples [24,25,26].

Exploratory factor analysis was performed by analyzing the main components, and for interpretation was considered Kaiser’s criterion (superunit value criterion), the choice of the number of axes for which the eigenvalues correspond to a value greater than one (eigen value). The value of the Kaiser–Meyer–Olkin index and the level of significance of the Bartlett sphericity test suggested the existence of one or more common factors and the application of a factor reduction. The analysis of the data after the oblique rotation of the variables correlated with the “Varimax” method allowed the extraction of the factors, the hierarchy of the distribution of the weights of each component and the interpretation of the results [26]. The *p* value ≤ 0.05 was considered statistically significant, and the statistical analysis was performed with the program S.P.S.S.-I.B.M. Statistics version 22.

## 3. Results

The analysis of the results was performed for 522 patients with glaucoma in the records of the Ophthalmology office within IOCECHO during the years 1999–2019. During this period, 150,844 people with ophthalmic pathology were consulted, from which 522 patients (0.34%) with the diagnosis of glaucoma were selected, of which: 184 people (35.2%) with POAG and 338 people (64.8%) with NTG.

According to the records in the current medical archive of the hospital, the annual proportion of cases diagnosed with POAG in the total number of patients examined was between 0.1% in 2005, 2008, and 2010 and 2.4% in 2012, when 101 people were detected. The annual average detection of people with glaucoma was 26 people/year, and in 2012 there was the highest number of people detected, 101 patients (19.3%). In the studied records, no cases of uni- and/or bilateral blindness were mentioned.

The distribution of patients in the study by gender recorded a total of: 140 men (26.8%) and 382 women (73.2%). The sex ratio (sex ratio—M/F) is 0.37. Statistically, the numerical composition of the male group differs significantly from the numerical composition of the female group (Binominal Test, Exact Sig. ≤ 0.05, 2-tailed, *p* = 0.000). Of the 522 patients with glaucoma, 403 (77.2%) live in urban areas, of which 372 people (71.3%) in Oradea (county capital) and 119 people (22.8%) in rural areas.

The distribution of the total number of glaucoma cases recorded in 2019, according to age group and gender shows an increasing frequency in both genders in the age group between 65–69 years in total 105 people (20.2%), of which 28 men (23.9%) and 204 women (76.1%) (Table 2.). The mean age of the patients was 60.81 years with DS ± 12.14 with a minimum of 20 years and a maximum of 90 years. In men, high frequencies are observed in the age groups: 65–69 years, 28 people (20.0%); in the age group 75–79 years 24 people (17.2%); and in the age group 70–74 years, 19 people (13.6%). In women, the distribution of high frequencies is recorded in the age groups: 65–69 years with 77 people (20.2%); 60–64 years with 74 people (19.4%); and in the age group 70–74 years, 53 people (13.9%).

The distribution of the number of cases at the “first consultation” by age group and gender, shows an increased frequency for men and women in the age groups between 55–69 years; a total of 272 people (52.1%), of which 66 men (24.3%) and 206 women (75.7%), (Table 3). The mean age of the patients was 60.81 ± 12.14 years with a minimum of 20 years and a maximum of 90 years. In men, high frequencies are observed in the age groups: 65–69 years, 26 people (18.5%) and equally for the age groups 55–59 years and 60–64 years (14.3%). In women, the distribution of high frequencies is recorded in the age group: 55–59 years with 81 people (21.4%); 60–64 years with 73 people (19.1%); and in the age group 65–69 years, 52 people (13.6%). Out of the total of 522 patients registered, 489 (93.7%) were new cases detected and 33 patients (6.3%) were cases with regular monitoring.

The mean age of the patients at the last consultation was 66.10 years with DS ± 12.47, with a minimum of 22 years and a maximum of 93 years (Table 4). The distribution of the number of patients who completed the monitoring and treatment procedure by age group and gender, shows the highest frequency in both genders in the age groups 65–69 years, as follows: 96 people (18.4%), of which 27 were men (28.1%) and 69 were women (71.9%). In men, at the last consultation, the highest frequency is recorded in the age group 65–69 years, with 27 people (19.2%), then equally in the age groups 55–59 years, 70–74 years, and 75–79 years for every 18 people (12.9%). In women, at the last consultation, the highest frequency is recorded in the age group 65–69 years with 69 people (18.1%), followed by the age groups 60–64 years: 66 people (17.2%) and the age group 70–74 years with 55 people (14.4%).

The annual average of people who completed the monitoring and treatment procedure for glaucoma was about 10 people/year (199 people/20 years), and in 2013 there was the highest number of people with abandonment, 41 patients (20.6%). The reasons for abandoning monitoring and treatment were: death in 21 people (10.5%) and voluntary cessation of treatment (low adherence) for 178 people (89.5%). The time interval between the initial and final consultation (monitoring and treatment period) had an average of 5.30 ± 3.47 years, with a minimum of 1 year and a maximum of 20 years. The multiannual distribution of the monitoring duration shows the predominance of the 1-year interval, in 94 people (25 men—26.59% and 69 women—73.41%). In Figure 1. it can be seen that more than half of the monitored patients, respectively 279 people (53.44%) are included in the first five years of medical supervision (77 men—27.59% and 202 women—72.40%).

Among the family medical history in the patient records in men “Diabetes mellitus” (first degree relatives) with a frequency of 6.4%, “Chronic glaucoma” (first degree relatives) with a frequency of 5.7%, and hypertension (first degree relatives) with a frequency of 4.3%. In women, the heredocolateral antecedents were: “Chronic glaucoma” (first degree relatives) with a frequency of 6.8% and Diabetes mellitus (first degree relatives) with a frequency of 4.3%. Pathological personal history in men includes: hypertension with a frequency of 15.7%, “chronic glaucoma” with a frequency of 8.6%, and “type II diabetes with a frequency of 6.4%; and in women: “Chronic glaucoma” with a frequency of 14.1%, “Type II diabetes with a frequency of 11.3%, and hypertension with a frequency of 7.9%.

The ophthalmological examination recorded the determination of IOP and CDR (Cup to disc ratio; cup/disc ratio). In patients with POAG, between the initial and the final consultation, IOP decreased statistically significantly by 46.16%, from 30.50 ± 7.98 mmHg to 16.42 ± 3.01 mmHg (*p* = 0.000). In patients with NTG, in between the initial and final consultation, the IOP decreased significantly by 17.44%, from 16.39 ± 3.66 mmHg to 13.53 ± 1.92 mmHg (*p* = 0.000). CDR for the whole group of patients recorded an increase from the first to the last consultation. In patients with POAG the increase was statistically significant from 0.58 ± 0.11 to 0.69 ± 0.10 (*p* = 0.000), and in patients with NTG, the increase was from 0.60 ± 0.23 to 0.70 ± 0.18 (without statistical significance *p* = 0.795). The ocular functional examination included the determination of visual acuity (VA) and analysis of VF parameters. At the time of diagnosis for patients with POAG, VA was ≥0.7 in 80 people (43.5%) and <0.7 in 104 people (56.5%), and in patients with NTG, VA was ≥ 0.7 in 93 people (27.5%) and <0.7 in 239 people (72.5%). At the last consultation for patients with POAG, VA was ≥0.7 in 72 people (39.15%) and <0.7 in 112 people (60.9%), and in patients with NTG, VA was ≥0.7 in 83 people (24.5%) and <0.7 in 255 people (75.5%).

The interpretation of the results at the computerized perimeter was performed by analyzing the credibility indices and the VF parameters. The average time (minutes) required to perform VF tests in glaucoma patients increased from the initial to the final consultation: at POAG it increased from 9.26 ± 2.66 min to 10.18 ± 2.40 min, statistically significant (*p* = 0.000), and in patients with NTG the duration of examination increased from 9.78 ± 2.53 min to 10.96 ± 2.06 min, statistically significant (*p* = 0.000). Among the credibility indices, the average scores of the “loss of fixation” indicator decreased between the first and last VF examination for both POAG and NTG, these indices are statistically significant (* *p* ≤ 0.05) at an average percentage threshold of maximum 16–20% (POAG *p* = 0.035; NTG *p* = 0.025). The mean scores for the “false positive” and “false negative” errors increased for both POAG and NTG, without statistical significance (* *p* ≤ 0.05) at the average percentage level of maximum ≤ 15% (Table 5).

The study of VF parameters between the first and last examination presents the following aspects (Table 6), the visual slope averages at 10 degrees for both POAG and NTG are statistically significant (POAG *p* = 0.002; NTG *p* = 0.000). The averages recorded for “Zero level” (*p* = 0.000), “Average” (*p* = 0.006), and “PD” (*p* = 0.001), are statistically significant only for NTG. In patients with POAG, the averages of the indicators are on a slightly upward trend without statistical significance. The “Level at 3 degrees” and “AD” averages for both POAG and NTG are without statistical significance (Table 6).

In assessing the visual function, the “Bebie Curve” graph was considered for the rapid assessment of the integrity of the visual field in relation to age (Table 7). In between the first and last examination in patients with POAG, there is an increase in the frequency of the type III chart (with small but profound defects on the CV) from 83 people (44%) to 106 people (56%) and a decrease in the frequency of the type chart. IV (VF with very extensive and shallow defect) from 80 people (42%) to 61 people (32%). In patients with NTG, the phenomenon is reversed, increasing the frequency of the type IV chart from 89 people (27%) to 92 people (28%) and a decrease in the frequency of the type III chart from 161 people (48%) to 149 people (45%).

The distribution of the centralized defect resulting from the VF (Bebie Curve) examination shows for both forms of disease for both POAG and NTG, from the first and last examination a predominance of the type III and IV model. The Chi-Square test evaluates the existence of a statistically significant difference between the frequencies of the “Bebie Curve” indicator types (*p* = 0.000). The correlation analysis was performed separately for each form of disease: POAG and NTG. The study of the correlations between the VF parameters in the patients with POAG, at the first consultation (C1) and at the last consultation (C2) was performed according to the indicators from Table 8. The vast majority of parameters have non-normal distribution.

The analysis of the correlations at moderate level between the VF parameters in patients with POAG, at the first consultation (C1) and at the last consultation (C2) have statistical significance (Table 9). The correlation between “Level 3 degrees—AD” identifies a direct statistical link at a moderate level between visual slope sensitivity and significant generalized defects on loss of eye sensitivity (calculated as the average deviation of the eye from the desirable profile, based on age). The correlations between “Zero level—Average” and “Level 3 degrees—Zero level” identify direct statistical links at moderate level with expression on retinal sensitivity. The correlation between “Average—PD” a close negative link between the average sensitivity of the retina and localized defects (reflects the amount and depth of local defects—scotomas). 

The study of the correlations between the VF parameters in the patients with NTG, at the first consultation (C1) and at the last consultation (C2) was performed according to the indicators from Table 10. The vast majority of parameters have non-normal distribution.

The analysis of the correlations at moderate level between the VF parameters in patients with NTG at the first consultation (C1) and at the last consultation (C2) have statistical significance (Table 10 and Table 11). Correlations between: “Zero level—Average”; “Level 3 degrees—Zero level”, “Level 3 degrees—Average” describe direct statistical links at moderate level with expression on retinal sensitivity. The “Level 3 degrees—AD” correlation identifies a moderately direct statistical link between visual slope sensitivity and significant generalized defects on loss of eye sensitivity (calculated as the mean of the eye deviation from the desirable age-based profile). The correlation between “Slope 10 degrees” and “Average” identifies a close negative link between visual slope sensitivity and average retinal sensitivity.

Exploratory factor analysis led to the extraction of factors that have eigenvalues equal to or equal to 1.00; following the application of the orthogonal rotation of the factors—Varimax (Table 12 and Table 13).

In the group of POAG patients, three factors were extracted at the first examination: “Level 30” (36.72%), “Average” (26.33%), and “PD” (20.13), which together cover 83.20% of the variation of the analyzed values, the remaining variation up to 100% remains unexplained by this factorial model. They characterized the sensitivity of the visual slope, the average sensitivity of the retina and localized defects. At the second examination: “Level 3 degrees” (55.72%) and “Slope at 100 (23.88%) were the extracted factors; these covering 79.60% of the variation of the analyzed values and express the sensitivity and inclination of the visual slope. In the group of NTG patients, two factors were extracted at the first examination: “Level 30” (57.38%) and “PD” (28.81), which together represent 86.19% of the variation of the analyzed values. They characterized the sensitivity of the visual slope, the average sensitivity of the retina, and localized defects. At the second examination: Average (42.24%) and “Level 3 degrees” (31.38%) were the extracted factors; covering 73.82% of the variation of the analyzed values expressing the average retinal sensitivity and visual slope sensitivity. 

The common element of the extracted factors is the sensitivity of the visual slope, which is the first affected element, with subsequent resonance on the visual function. The recommended and followed treatment for POAG and NTG patients is summarized in Table 14 and Table 15.

## 4. Discussion

Glaucoma is a chronic disease, with progressive and silent evolution, in most cases being undiagnosed until advanced stages due to the asymptomatic nature of the disease in its early forms. As the treatment is long-lasting, it must be adapted and constantly reviewed.

A similar study, based on the analysis of medical records, reported for a period of 2 years (January 2003–January 2005), a proportion of POAG detected in the total outpatient consultations of 0.74% (827 cases) to a number of 110,794 people investigated, with an annual average detection of approximately 413 people/year [27]. Another research presents the proportion of POAG detection from the total outpatient consultations as outpatient as 1.37%, for 129 patients at a number of 9406 consultations, over a period of three years: 2007–2009, with an annual average of 43 people/year. It also mentions the alteration of specific ophthalmological parameters [28].

The results of our research mention 522 people (0.35%) registered and diagnosed with POAG, out of a total of 150,844 patients with ophthalmological pathologies, consulted in the period 1999–2019, with an annual average detection of approximately 26 people/year. A special importance in the specialized medical practice is the prevention activities. Secondary prevention of POAG is a current topic, its purpose being: early detection, institution of treatment, reducing the progression of asymptomatic disease, and improving the quality of life and in certain forms of “check-up” (on-demand health check-up, regular medical check-up in a specialized service) [12,29]. 

Screening programs for high-risk glaucoma population groups are more effective compared to population screening that has not been shown to be cost-effective [30,31]. The detection of POAG is based on clinical and paraclinical examination (especially on the determination of IOP, VF examination and other current investigations) highlighting the risk of the disease and the implications for public health policy and planning [32].

IOP is the only modifiable risk factor in patients with high pressure glaucoma (high intraocular pressure is correlated, clinically, and paraclinically with POAG), and its decrease prevents disease progression, emphasizing the importance of early diagnosis and treatment [28,33,34,35,36,37,38,39]. Clinical practice has shown that adequate reduction in IOP remains the main element of patient management, being an effective intervention, regardless of the subtype or stage of the disease (POAG and NTG) [35,40,41].

Visual field testing supplemented with new techniques provides information that can detect disease progression and improves clinical decision-making, being useful for monitoring patients with advanced glaucoma and reducing the burden of the disease [42,43]. Routine analysis of VF parameters performed with SAP technique (“Standard Automated Perimetry”) is the functional methodology with the most validated results [44,45,46,47,48,49,50].

Although the VF testing technique has not changed substantially for about 150 years, it will continue to play an important role in the diagnosis and management of glaucoma, being considered the “gold standard”. Progress is constantly being made in test administration, standardization, statistical evaluation, clinical analysis, and interpretation and prediction of the result based on the findings of the VF examination [51,52]. The only effective treatment to slow the progression of POAG is to reduce intraocular pressure (topical treatment, anti-glaucoma eye drops, and laser or surgical treatments), which acts on this main risk factor [53]. 

Early treatment with an individualized approach requires active monitoring and more rigorous medical therapy for IOP. Glaucoma treatment is a continuous post-diagnosis process throughout the entire life [54,55,56,57,58]. Many specialized studies have highlighted the important role of tertiary prevention of POAG, by providing specialized care that ensures effective control of IOP and implicitly of the long-term evolution of the disease [58]. Glaucoma should not be managed in isolated manner; the aim of the actions is to detect and treat all potential causes of blindness with a focus on preventive actions. The routine of full examination, monitoring and treatment becomes mandatory, in order to reduce the economic and social costs generated by the disease [59,60]. Drug treatment must be specific and constantly monitored [61,62]. Clinicians must ensure that patients remain adherent to the administration of glaucoma medications and must monitor for adverse events in medical or surgical interventions used to treat glaucoma [63,64,65]. 

The strengths of the study are the large number of subjects included in the analysis, the retrospective nature of the study, the long duration of the investigation (20 years) and the use of only the data recorded in the official records in the medical archive (individual consultation sheets with the printed visual field examination). The limits of the study related to the design of the initial medical records (1999), the examination of patients with tools and methods available at that time, as well as the lack of modern updates for high-performance paraclinical ophthalmic investigations.

## 5. Conclusions

Tertiary prevention of glaucoma, by providing comprehensive specialist care, ensures effective control of IOP and implicitly of the long-term evolution of the disease. Intraocular pressure (IOP) is the only modifiable risk factor in glaucoma patients (intraocular pressure being correlated, clinically and paraclinically with POAG) and its decrease prevents the progression of the disease, emphasizes the importance of early diagnosis and treatment. Monitoring and treatment of patients with glaucoma was beneficial, a fact confirmed by the improvement of the main risk factor: intraocular pressure, which decreased statistically significantly: in patients with POAG by 46.16%; from 30.50 ± 7.98 mmHg to 16.42 ± 3.01 mmHg (*p* = 0.000) and in those with NTG by 17.44% from 16.39 ± 3.66 mmHg to 13.53 ± 1.92 mmHG (*p* = 0.000). Management of glaucoma patients is very important and consists of: complete eye examination, visual field testing (a valuable tool for eye diagnosis), complete with new techniques that can more effectively detect disease progression and improve therapeutic decision making. The duration of treatment and monitoring of glaucoma patients was on average 5.1 ± 3.4 years, for 184 patients (35.2%) with POAG and 5.1 ± 3.8 years, for 338 patients (64.8%) with NTG.

Application of complex and scientific treatments can have a major impact on the evolution of glaucomatous optic neuropathy, and specific structural and functional paraclinical investigations facilitate an efficient detection, treatment and monitoring. The optimization of visual functions, the treatment and decrease of the prevalence of POAG and other ophthalmological pathologies, can have an important impact on the quality of life of patients. The treatment of POAG can decrease deterioration of visual function and have an important impact on the quality of life of patients.

## Figures and Tables

**Figure 1 jpm-11-00830-f001:**
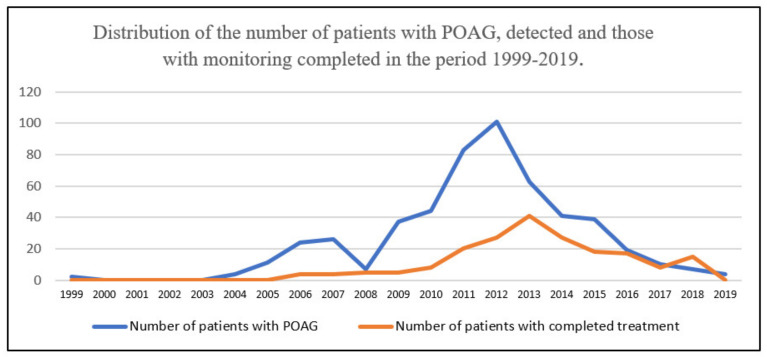
Distribution of the number of POAG patients detected and patients with treatment completed in the period 1999–2019.

**Table 1 jpm-11-00830-t001:** Classification of the centralized defect of a VF test result (Bebie Curve).

Bebie Curve type I	Extensive and deep damage to the “visual field”
Bebie Curve type II	No real defects in the “Field of View”
Bebie Curve type III	Small but deep defects of “Field of View”
Bebie Curve type IV	A “Field of View” with a very large and shallow defect

VF—visual field.

**Table 2 jpm-11-00830-t002:** Distribution of patients with POAG by gender and age group in 2019.

Age Group	Male Group POAG		Female Group POAG		Whole Group POAG	
	N	%	N	%	N	%
20–24	0	0.0	1	0.3	1	0.2
25–29	0	0.0	4	1.0	4	0.8
30–34	1	0.7	1	0.3	2	0.4
35–39	0	0.0	4	1.0	4	0.8
40–44	3	2.1	3	0.8	6	1.1
45–49	5	3.6	13	3.4	18	3.4
50–54	9	6.4	26	6.8	35	6.7
55–59	7	5.0	25	6.5	32	6.1
60–64	17	12.1	74	19.4	91	17.4
65–69	28	20.0	77	20.2	105	20.2
70–74	19	13.6	53	13.9	72	13.8
75–79	24	17.2	42	11.0	66	12.6
80–84	14	10.0	33	8.6	47	9.0
over 85	13	9.3	26	6.8	39	7.5
Total	140	100	382	100	522	100

POAG—primary open-angle glaucoma with high intra-ocular pressure; N—number of cases.

**Table 3 jpm-11-00830-t003:** Distribution of glaucoma patients by gender and age group at the first consultation.

Age Group	Male Group POAG		Female Group POAG		Whole Group POAG	
	N	%	N	%	N	%
20–24	1	0.7	5	1.3	6	1.1
25–29	1	0.7	4	1.0	5	1.0
30–34	0	0.0	4	1.0	4	0.8
35–39	4	2.9	3	0.8	7	1.3
40–44	6	4.3	18	4.7	24	4.6
45–49	11	7.9	27	7.1	38	7.3
50–54	9	6.4	36	9.4	45	8.6
55–59	20	14.3	81	21.4	101	19.3
60–64	20	14.3	73	19.1	93	17.9
65–69	26	18.5	52	13.6	78	14.9
70–74	19	13.6	36	9.4	55	10.5
75–79	13	9.3	28	7.3	41	7.9
80–84	9	6.4	8	2.1	17	3.3
over 85	1	0.7	7	1.8	8	1.5
Total	140	100	382	100	522.0	100

POAG—primary open-angle glaucoma with high intra-ocular pressure; N—number of cases.

**Table 4 jpm-11-00830-t004:** Distribution of glaucoma patients by gender and age group at the last consultation.

Age Group	Male Group POAG		Female Group POAG		Whole Group POAG	
	N	%	N	%	N	%
20–24	0	0.0	2	0.5	2	0.4
25–29	0	0.0	3	0.8	3	0.6
30–34	1	0.7	1	0.3	2	0.4
35–39	2	1.4	5	1.3	7	1.3
40–44	3	2.1	6	1.6	9	1.7
45–49	6	4.3	19	5.0	25	4.8
50–54	7	5	22	5.8	29	5.6
55–59	18	12.9	51	13.4	69	13.2
60–64	12	8.6	66	17.1	78	14.9
65–69	27	19.2	69	18.1	96	18.4
70–74	18	12.9	55	14.4	73	14.0
75–79	18	12.9	37	9.7	55	10.5
80–84	16	11.4	20	5.2	36	6.9
over 85	12	8.6	26	6.8	38	7.3
Total	140	100	382	100	522	100

POAG—primary open-angle glaucoma with high intra-ocular pressure); N—number of cases.

**Table 5 jpm-11-00830-t005:** Distribution of credibility indices in the interpretation of the “visual field”.

Indicators	Initial Consultation	Final Consultation	Z	*p **	Considered Values
Loss of Fixation—POAG	15.56 ± 4.28	13.44 ± 3.87	−2.10	0.035	16–20% = average
Loss of Fixation—NTG	14.14 ± 4.31	12.55 ± 4.54	−2.24	0.025	16–20% = average
False positive—POAG	8.54 ± 3.64	9.23 ± 3.05	−0.66	0.503	≤15%
False positive—NTG	9.88 ± 3.33	10.64 ± 2.81	−0.91	0.361	≤15%
False negative—POAG	11.27 ± 3.40	12.44 ± 2.20	−1.19	0.231	≤15%
False negative—NTG	11.29 ± 2.55	12.13 ± 2.41	−1.30	0.192	≤15%

POAG—primary open-angle glaucoma with high intra-ocular pressure); NTG primary open-angle glaucoma with normal intra-ocular pressure, particular form of open-angle glaucoma); Z—Two-Related Samples—Wilcoxon test; *p **—level of statistical probability ≤ 0.05.

**Table 6 jpm-11-00830-t006:** Distribution of the parameters “Visual field” of the patient with POAG and NTG.

Parameter	Initial Consultation	Final Consultation	Z	*p **
Level at 3°—POAG	27.95 ± 5.91	28.10 ± 4.15	−0.26	0.794
Level at 3°—NTG	28.24 ± 6.83	30.11 ± 4.88	−0.71	0.474
Visual slope at 10°—POAG	2.42± 0.99	2.21 ± 0.96	−3.13	0.002
Visual slope at 10°—NTG	2.38 ± 1.09	1.81 ± 0.91	−4.9	0.000
Zero Level—POAG	20.85 ± 4.71	22.28 ± 5.99	−1.7	0.085
Zero Level—NTG	21.08 ± 6.06	24.70 ± 4.41	−4.5	0.000
Average—POAG	16.15 ± 11.73	23.01 ± 6.75	−1.95	0.056
Average—NTG	18.55 ± 11.02	22.49 ± 7.85	−2.74	0.006
PD—POAG	3.40 ± 3.27	3.78 ± 2.98	−1.0	0.315
PD—NTG	2.47 ± 3.10	3.03 ± 2.85	−3.27	0.001
AD—POAG	−0.27 ± 5.12	0.17 ± 4.81	−0.46	0.644
AD—NTG	−0.28 ± 3.98	0.55 ± 5.07	−1.31	0.190

POAG—primary open-angle glaucoma with high intra-ocular pressure; NTG—primary open-angle glaucoma with normal intra-ocular pressure, particular form of open-angle glaucoma); AD—average defect; PD—pattern deviation; *p ** —level of statistical probability ≤ 0.05.

**Table 7 jpm-11-00830-t007:** Distribution of the types of “Bebie Curve” indicators of the centralized defect resulting from the examination of the “visual field” for patients with POAG and NTG.

Model, Bebie Chart	C1-POAG		C2-POAG		C1-NTG		C2-NTG	
	CA	%	CA	%	CA	%	CA	%
Bebie curve type I	5	2	9	5	33	10	38	11
Bebie curve type II	22	12	14	7	49	15	53	16
Bebie curve type III	83	44	106	56	161	48	149	45
Bebie curve type IV	80	42	61	32	89	27	92	28
Total	190	100	190	100	332	100	332	100

POAG—primary open-angle glaucoma with high intra-ocular pressure; NTG—primary open-angle glaucoma with normal intra-ocular pressure, particular form of open-angle glaucoma; C1 = visual field—initial consultation; C2 = visual field—final consultation.

**Table 8 jpm-11-00830-t008:** Comparative distribution of correlations between “Visual Field” parameters in patients with POAG at the first consultation (VF1) and at the last consultation (VF2).

VF Parameter	*p* ***	rs/*prs*	Slope 10°	Level 3°	Zero Level	Average	PD	AD
C1-Slope 10° POAG	0.269	rs	1.000	−0.008	−0.284 *	−0. 454 **	−0.165	−0.052
		*p*	-	0.947	0.021	0.002	0.077	0.615
C2-Slope 10° POAG	0.482	rs	1.000	0.284	−0.275 *	−0.217	−0.191	−0.059
		*p*	-	0.088	0.032	0.178	0.150	0.667
C1 Level 3° POAG	0.038	rs	−0.008	1.000	0.662 **	0.534 **	−0.441 **	0.757 **
		*p*	0.947	-	0.000	0.001	0.000	0.000
C2 Level 3° POAG	0.105	rs	0.284	1.000	0.718 **	0.675 **	−0.262	0.797 **
		*p*	0.088	-	0.000	0.000	0.117	0.000
C1 Zero Level POAG	0.016	rs	−0.284 *	0.662 **	1.000	0.786 **	0.528 **	0.097
		*p*	0.021	0.000	-	0.000	0.000	0.560
C2 Zero Level POAG	0.572	rs	−0.275 *	0.718 **	1.000	0.639 **	0.405 **	0.510 **
		*p*	0.032	0.000	-	0.000	0.002	0.000
C1 Average POAG	0.102	rs	−0.454 **	0.534 **	0.786 **	1.000	0.154	0.471 **
		*p*	0.002	0.001	0.000	-	0.319	0.005
C2 Average POAG	0.080	rs	−0.217	0.675 **	0.639 **	1.000	−0.742 **	0.686 **
		*p*	0.178	0.000	0.000	-	0.000	0.000
C1—PD POAG	0.001	rs	−0.165	−0.441 **	0.528 **	0.154	1. 000	−0.265 **
		*p*	0.077	0.000	0.000	0.319	-	0.009
C2—PD POAG	0.018	rs	−0.191	−0.262	0.405 **	−0.742 **	1. 000	−0.145
		*p*	0.150	0.117	0.002	0.000	-	0.286
C1—AD POAG	0.000	rs	0.052	0.757 **	0.097	0.471 **	−0.265 **	1.000
		*p*	0.615	0.000	0.560	0.005	0.009	-
C2—AD POAG	0.001	rs	−0.059	0.797 **	0.510 **	0.686 **	−0.145	1.000
		*p*	0.667	0.000	0.000	0.000	0.286	-

POAG -primary open-angle glaucoma with high intra-ocular pressure; NTG—primary open-angle glaucoma with normal intra-ocular pressure, particular form of open-angle glaucoma; r_s_—Spearman’s rho coefficient **. Correlation is significant at the *p* = 0.01 level (2-tailed); Spearman’s rho *. Correlation is significant at the *p* = 0.05 level (2-tailed); C1 = visual field—initial consultation; C2 = visual field—final consultation; *** Kolmogorov-Smirnov *p*-value.

**Table 9 jpm-11-00830-t009:** Distribution of moderate-intensity correlations between “Visual Field” parameters in patients with POAG, at the first consultation (C1) and at the last consultation (C2).

Nr	POAG Correlation	C1	C2	Type of Correlation		Intensity Level	r_s_	*p*
				Pozitiv	Negativ			
1.	”Level 3°—AD”	0	1	1	0	moderat	0.797 **	0.000
2.	”Level 3°—AD”	1	0	1	0	moderat	0.757 **	0.000
3.	”Zero Level—average”	1	0	1	0	moderat	0.787 **	0.000
4.	”Level 3°—Zero level”	0	1	1	0	moderat	0.718 **	0.000
5.	”Average—PD”	0	1	0	1	moderat	−0.742 **	0.000

r_s_—Spearman’s rho coefficient **. Correlation is significant at the *p* = 0.01 level (2-tailed); Spearman’s rho; C1-visual field—initial consultation; C2-visual field—final consultation, AD—average defect; PD—pattern deviation.

**Table 10 jpm-11-00830-t010:** Comparative distribution of “Visual field” parameters in patients with POAG at the first consultation (C1) and at the last consultation (C2).

VF Parameter	*p* ***	rs/*p*	Slope 10°	Level 3°	Zero Level	Average	PD	AD
C1-Slope 10° NTG	0.011	rs	1.000	0.098	−0.475 **	−0.539 **	−0.230 **	−0.210 **
		*p*	.	0.255	0.000	0.000	0.000	0.003
C2-Slope 10° NTG	0.000	rs	1.000	0.242 *	−0.406 **	−0.193	0.039	0.114
		*p*	.	0.024	0.000	0.059	0.680	0.241
C1 Level 3° NTG	0.004	rs	0.098	1.000	0.825 **	0.769 **	−0.142	0.717 **
		*p*	0.255	.	0.000	0.000	0.098	0.000
C2 Level 3° NTG	0.095	rs	0.242 *	1.000	0.587 **	0.664 **	−0.134	0.698 **
		*p*	0.024	.	0.000	0.000	0.217	0.000
C1 Zero Level NTG	0.001	rs	−0.475 **	0.825 **	1.000	0.888 **	0.444 **	00.581 **
		*p*	0.000	0.000	.	0.000	0.000	0.000
C2 Zero Level NTG	0.033	rs	−0.406 **	0.587 **	1.000	0.853 **	−0.093	0.231 *
		*p*	0.000	0.000	.	0.000	0.345	0.021
C1 Average NTG	0.000	rs	−0.539 **	0.769 **	0.888 **	1.000	0.127	0.650 **
		*p*	0.000	0.000	0.000	.	0.092	0.000
C2 Average NTG	0.000	rs	−0.193	0.664 **	0.853 **	1.000	−0.397 **	0.427 **
		*p*	0.059	0.000	0.000	.	0.000	0.000
C1—PD NTG	0.000	rs	−0.230 **	−0.142	0.444 **	0.127	1.000	−0.114
		*p*	0.000	0.098	0.000	0.092	.	0.114
C2—PD NTG	0.000	rs	0.039	−0.134	−0.093	−0.397 **	1.000	−0.130
		*p*	0.680	0.217	0.345	0.000	.	0.179
C1—AD NTG	0.000	rs	−0.210 **	0.717 **	0.581 **	0.650 **	−0.114	1.000
		*p*	0.003	0.000	0.000	0.000	0.114	.
C2—AD NTG	0.139	rs	0.114	0.698 **	0.231 *	0.427 **	−0.130	1.000
		*p*	0.241	0.000	0.021	0.000	0.179	.

POAG—primary open-angle glaucoma with high intra-ocular pressure); NTG—primary open-angle glaucoma with normal intra-ocular pressure, particular form of open-angle glaucoma; **. Correlation is significant at the *p* = 0.01 level (2-tailed); *. Correlation is significant at the *p* = 0.05 level (2-tailed).; C1—visual field—initial consultation; C2—visual field—final consultation, *** Kolmogorov-Smirnov *p*-value, AD—average defect; PD—pattern deviation.

**Table 11 jpm-11-00830-t011:** Distribution of moderate-intensity correlations between “Visual Field” parameters in patients with NTG, at the first consultation (C1) and at the last consultation (C2).

No	Correlation	C1	C2	Type of Correlation		Intensity Level	r_s_	*p*
				Pozitiv	Negativ			
1.	”Zero level—Average”	1	0	1	0	strong	0.888 **	0.000
2.	”Zero level—Average”	0	1	1	0	strong	0.853 **	0.000
3.	”Level 3°—Zero level”	1	0	1	0	strong	0.825 **	0.000
4.	”Level 3°—Zero level”	0	1	1	0	moderate	0.587 **	0.000
5.	”Level 3°—Average”	1	0	1	0	moderate	0.769 **	0.000
6.	”Level 3°—Average”	0	1	1	0	moderate	0.664 **	0.000
7.	”Level 3°—AD”	1	0	1	0	moderate	0.717 **	0.000
8.	”Level 3°—AD”	0	1	1	0	moderate	0.698 **	0.000
9.	”Slope 10°—Average”	1	0	0	1	moderate	−0.539 **	0.000

r_s_—Spearman’s rho coefficient **. Correlation is significant at the *p* = 0.01 level (2-tailed); Spearman’s rho; C1—visual field—initial consultation; C2—visual field—final consultation; AD—average defect; PD—pattern deviation.

**Table 12 jpm-11-00830-t012:** Comparative distribution of the common factors of the “Visual field” indicators of patients with POAG at the first and second consultation (C1 and C2).

	Rotated Component Matrix									
	Factor 1			Factor 2			Factor 3			
	Parameter	Comp.	Variation %	Parameter	Comp.	Variation %	Parameter	Comp.	Variation %	Cumulated Variation %
CV1 POAG	Level 3°	0.953	36.72	Average	0.862	26.33	PD	0.950	20.13	83.20
CV2 POAG	Level 3°	0.951	55.72	Slope 10°	0.797	23.88				79.60
	Zero Level	0.899		PD	0.768					
	AD	0.869								
	Average	0.856								

POAG—primary open-angle glaucoma with high intra-ocular pressure; NTG—primary open-angle glaucoma with normal intra-ocular pressure, particular form of open-angle glaucoma; CV1—visual field—initial consultation; CV2—visual field—final consultation; —average defect; PD—pattern deviation.

**Table 13 jpm-11-00830-t013:** Comparative distribution of the common factors of the “Visual field” indicators of patients with NTG at the first and second consultation (C1 and C2).

	Rotated Component Matrix						
	Factor 1			Factor 2			
	Parameter	Comp.	Variation %	Parameter	Comp.	Variation %	Cumulated Variation %
CV1 NTG	Level 3°	0.946	57.38	PD	0.895	28.81	86.19
	Zero Level	0.937		Slope10 gr	0.895		
	Average	0.923					
	AD	0.904					
CV2 NTG	Average	0.911	42.24	Level 3°	0.876	31.58	73.82
	Zero Level	0.866		AD	0.861		
				Slope10 gr	0.554		

POAG—primary open-angle glaucoma with high intra-ocular pressure; NTG—primary open-angle glaucoma with normal intra-ocular pressure, particular form of open-angle glaucoma; CV1—visual field—initial consultation; CV2—visual field—final consultation; —average defect; PD—pattern deviation.

**Table 14 jpm-11-00830-t014:** Distribution of recommended drug classes and their dosage in the treatment of patients with POAG.

Anti-Glaucoma Drugs	Dosage	Male Group		Female Group		Total Group	
		N	%	N	%	N	%
Prostaglandin analogs	1 drop/day	28	50.0	95	74.2	123	66.8
Selective Beta blockers	2 × 2 drops/day	5	8.9	4	3.1	9	4.9
Beta blockers + Acetazolamid Inhibitors	3 × 1 drops/day	5	8.9	9	7.0	14	7.6
Prostaglandin analogs + Beta blockers	2 × 1 drops/day	14	25.0	20	15.6	34	18.5
Brizolamid 10 mg + timolol	2 × 1 drops/day	2	3.6	0	0.0	2	1.1
No mentioned treatment	-	2	3.6	0	0.0	2	1.1
Total		56	100.0	128	100.0	184	100.0

POAG—primary open-angle glaucoma with high intra-ocular pressure; N—active cases.

**Table 15 jpm-11-00830-t015:** Distribution of recommended drug classes and their dosage in the treatment of patients with calculated NTG.

Anti-Glaucoma Drugs	Dosage	Male Group		Female Group		Total Group	
		N	%	N	%	N	%
Prostaglandin analogs	1 drop/day	47	56.0	178	70.1	225	66.6
Selective Beta blockers	2 × 2 drops/day	9	10.7	19	7.5	28	8.3
Beta blockers + Acetazolamid Inhibitors	3 × 1 drops/day	7	8.3	18	7.1	25	7.4
Prostaglandin Analogs + Beta blockers	2 × 1 drops/day	21	25.0	39	15.4	60	17.8
Brizolamid 10 mg + timolol	2 × 1 drops/day	0	0.0	0	0.0	0	0.0
No mentioned treatment	-	0	0.0	0	0.0	0	0.0
Total		84	100.0	254	100.0	338	100.0

NTG—primary open-angle glaucoma with normal intra-ocular pressure, particular form of open-angle glaucoma; N—active cases.

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
