# Peer review of "Aspects of Tertiary Prevention in Patients with Primary Open Angle Glaucoma"

_jpm, 2021, doi:10.3390/jpm11090830_

Round 1

Reviewer 1 Report

There are too many apprterm is approximations and the english is defectuous. For exemple, "hole group" instead 'whole groupe".

Authors used "GPUD", but the usual term is "POAG" and GPN is better than GTN

Data on GTN/GPN are present twice, worded in the same way 

Does the visual field  been validated internationally?

Line 111: "the walls of the eyeball"=> do you mean "the retina"?

Line 112: "straight muscle" => improper translation

Line 117: the physiological action of oblique muscles is different now

Frequency of POAG is very low and the number of new cases is strange.

Author Response

Dear Reviewer

I really appreciate your recommendations. We tried, based on all the recommendations of the reviewers, to improve and correct the article.

I take into consideration the recommendations and we improved all the section of article.

Thank you very much for your help.

Reviewer 2 Report

The authors present the results of a very interesting observational study in which they examined the tertiary prevention of 522 patients with primary glaucoma.  The study is novel, as I can find no other similar studies other than a short-term study that the authors discuss.

The background and discussion are very well done, other than the need for some minor English editing.

The Methods & Results can be hard to follow, but they are salvageable with some attention to detail by the authors and -- perhaps -- an English editing service.

Here are some specific recommendations:

  1. Overall, prevention is the preferred term (instead of prophylaxis).
  2. Also, primary is a better term than primitive.
  3. Line 59: Please provide a reference for the statement that GLC is the leading wolrldwide cause of irreversible blindness.
  4. The term GPUD, GPUI are not common. Perhaps the authors should stick with more standard terms (i.e., primary open angle glc = POAG, low tension GLC = NTG, etc.)
  5. Line 90: Please edit to read - "..., as vision loss cannot be..."
  6. Lines 110-121 are not necessary nor pertinent to the study. Please remove.
  7. Line 133: Please difine the acronym AISSCJUO.
  8. Line 149: As mentioned, stick with one set of acronyms. Here the authors use "POAG", which I believe is more appropriate.
  9. Line 194: the correct spelling is "pattern"
  10. Table 1: Please remove the term "happy trigger" from this table.
  11. Line 195: Please define the Bebie curve. Simply, it is a custom feature of the Octopus brand of VF analyzers.  Regarding Bebie curves, there are multiple instances of the authors using the term "Baby" instead. Please edit.
  12. Table 2, Row 1: The word is "Whole" (not "Hole")
  13. All tables: Please clean up the legends. They are complete, but they are inconsistent. For examples, for Table 2, "GPUD (primitive open-angle glaucoma with high intra-ocular pressure); CA –active cases" should be eidted to be "GPUD = primitive open-angle glaucoma with high intra-ocular pressure); CA = active cases"
  14. Fig. 1: I do not see how the linear  relationship curve has any value. I suggest removing the linear curve form graph and legend.
  15. Line 315: By "moitoring sheets", do the authors mean "progress sheets", "patient records" or "care sheets"? Please clarify.
  16. Line 334: I believe "VA" is a better acronym than "AV"  AV means something different to eyecare providers.
  17. Table 5: Please edit column 1 to read "positive" and "negative" (instead of positiv and negativ).
  18. Line 395: The authors state that most of the distributions are not normal. I suggest that the header for Table 8 & 10, Column two read "K-S p-value" instead of just p***.  Then define "K-S p-value = p-level of Kolmogorov-Smirnov test for normality (p < .05 indicates non-normal)" in the legend.
  19. Line 407: please add space between VF and parameters.
  20. Table 14: I recommend dosage (instead of posology) as the more common term.
  21. Lines 516-517: Could the authors report the incidence RATE (522/150,844 = 3.46 per 1000) here instead of the incident number per year?

Thank you for allowing me to read and review this work.

Author Response

(The authors gave the same response as above.)

Reviewer 3 Report

By using the records of the ophthalmology office of AISSCJUO, this study tends to describe the healthcare situation of POAG in a county. Overall, the study is comprehensive and meaningful.

Please see a few comments below:

  1. Please describe in detail how many patients are covered in the database being used. Is it a county hospital? How many populations are in this county?
  2. Table 2-4, the last column should be ‘whole’, instead of ‘hole’.
  3. How reliable is the VF results? What is the format of VF results archived in the database? Is it pure text? How were the results extracted for the use of the study?

Author Response

(The authors gave the same response as above.)

Round 2

Reviewer 1 Report

The comments have improved the text that can be published as itis